# Mixed Electronic-Ionic Conductivity and Stability of Spark Plasma Sintered Graphene-Augmented Alumina Nanofibres Doped Yttria Stabilized Zirconia GAlN/YSZ Composites

**DOI:** 10.3390/ma16020618

**Published:** 2023-01-09

**Authors:** Olga Kurapova, Oleg Glumov, Ivan Smirnov, Yaroslav Konakov, Vladimir Konakov

**Affiliations:** 1Institute of Chemistry, Department of Physical Chemistry, Saint Petersburg State University, Universitetskya nab 7/9, 199034 St. Petersburg, Russia; 2Institute of Problems of Mechanical Engineering, V.O., Bolshoj pr., 61, 199178 St. Petersburg, Russia

**Keywords:** yttria stabilized zirconia, graphene-augmented γ-Al_2_O_3_ nanofibres, spark plasma sintering, nanocomposite, mixed electronic-ionic conductivity, impedance spectroscopy

## Abstract

Graphene-doped ceramic composites with mixed electronic-ionic conductivity are currently attracting attention for their application in electrochemical devices, in particular membranes for solid electrolyte fuel cells with no necessity to use the current collector. In this work, composites of the Y_2_O_3_-ZrO_2_ matrix with graphene-augmented γ-Al_2_O_3_ nanofibres (GAlN) were spark plasma sintered. The conductivity and electrical stability in cyclic experiments were tested using electrical impedance spectroscopy. Composites with 0.5 and 1 wt.% GAlN show high ionic conductivity of 10^−2^–10^−3^ S/cm at 773 K. Around 3 wt.% GAlN percolation threshold was achieved and a gradual increase of electronic conductivity from ~10^−2^ to 4 × 10^−2^ S/cm with an activation energy of 0.2 eV was observed from 298 to 773 K while ionic conductivity was maintained at elevated temperatures. The investigation of the evolution of conductivity was performed at 298–973 K. Besides, the composites with 1–3 wt.% of GAlN addition show a remarkable hardness of 14.9–15.8 GPa due to ZrC formation on the surfaces of the materials.

## 1. Introduction

Since the discovery of single layer graphene (Gr) [1], the ceramic matrix composites reinforced by Gr have attracted great attention. Over the last ten years, the use of graphene and its derivatives for the design of ceramic composites opened new frontiers for the applications of many existing ceramic matrices (SiC, Si_3_N_4_, ZrO_2_) as for example, energy conversion and storage, biomaterials and bone engineering, thermoelectric generators ([2,3], additive manufacturing technologies, bioceramics and bone regeneration engineering [4], aerospace and engineering, etc. [5,6,7,8].

Among them, Gr-doped cubic yttria-stabilized zirconia (YSZ) composites have gained a special interest [9]. The YSZ ceramics is well-known oxygen ionic conductor (up to 0.1 S/cm at 1273 K), commercially used in lambda sensors, and as pilot-scale electrochemical devices as high-temperature solid oxide fuel cells, and oxygen pumps. However long exploitation of the ceramics at about 973–1273 K results in the grain growth, recrystallization and degradation of ceramics. That is why there is a constant search for novel pathways to refine ceramics’ structure or decrease the temperature of the exploitation of the ceramics with no considerable loss in conductivity. As it is mentioned by D. Marinha in [10] the combination of the ionic conductivity of YSZ with the high electronic conductivity of graphene (the mobility of charge transfer carriers 1.5 × 10^4^ cm^2^/V × s) in a composite could appear as an alternative material for high and intermediate temperature electrochemical applications. Indeed, the composite possessing mixed electronic-ionic conductivity can be used as an anode and even an electrolyte membrane in SOFC with no need to use a current collector [11]. The most widely used materials for SOFC anodes are various Ni cermets [12] or mixed perovskites [13] that suffer from a number of significant drawbacks such as higher resistance, significant grain growth, sulfur poisoning and significantly different thermal expansion coefficients (TEC) comparing to the electrolyte [14,15,16]. The use of zirconia-graphene could help to avoid TEC mismatch between an anode and an electrolyte and increase the performance of the device [17]. In order to match the requirements to be used in SOFC the composites should fit a number of requirements listed in [11,18]. Namely, the difference between the ionic and electronic components should not be more than half to one order of magnitude.

To date, considerable progress has been achieved in the field of graphene/YSZ composites having improved electrical and thermal conductivity [9,10,19,20,21] and enhanced mechanical properties [22,23,24]. The percolation threshold was achieved for the tetragonal zirconia matrix upon the introduction of 1 wt.% graphene additive [20], whereas for cubic zirconia upon the addition of 1 wt.% graphene additive. However, in all cases the electronic conductivity significantly predominated over the ionic component or a gradual decrease in stability was reported. As known the major problems associated with graphene-doped materials are (i) the difficulty in homogeneous Gr distribution in the composite matrix and (ii) the damaging of Gr in the matrix during milling and sintering steps [5,22]. In order to overcome mentioned problems, various graphene derivatives are used nowadays for high-performance composites manufacturing instead of pristine Gr, such as graphene oxide (GO) [25,26], reduced graphene oxide (rGO) [18,21,27,28], graphene nanoplatelets (GNP) [3,10,23], etc. They have a different number of stacked graphene layers, edge functional groups, alternating lateral sizes of a sheet etc., which enhance the bonding between the matrix and the additive and thus improve the properties of resulting composites [29]. Among them, rGO appears to be the most commonly used [27,29,30,31]. The structure of rGO is similar to pristine graphene, but contains residual functional groups and structural defects at the edge, which makes it more stable towards agglomeration in the metallic and ceramic matrices. Also, in the series of works [9,18,21,26,32,33] the effect of the sintering and shaping conditions on the structure and the properties of rGO/YSZ composites was studied. Based on the dilatometry data authors of [32] have shown that graphene-zirconia composites demonstrate the mixed mechanism of sintering with a grain boundary diffusion predominance. Such techniques as spark plasma sintering (SPS) [9,10,19,33], sintering using silicon carbide bed [21], sintering in the air [18], tape casting [26] and hot pressing [33] are used in the literature to produce zirconia/graphene composites. Following [33] SPS and hot-pressing techniques were shown to be effective to produce fully dense zirconia/carbon nanofiber composites with high electronic conductivity. The approaches, which included sintering in the air or using SiC powder bed did not allow for preserving graphene phase in the composites without its structural degradation, whereas SPS turned out to be successful for the task.

Recently, such graphene-containing modifier as graphene-augmented γ-Al_2_O_3_ nanofibres (GAlN), was suggested to improve the mechanical and thermal properties of alumina matrix [34]. When compared to Al_2_O_3_ the composites doped with 1–5 wt.% GAlN exhibited up to 90% higher wear resistance under severe conditions along with a decrease in the coefficient of friction only for the composite added by 1 wt.% of fibres. The γ-Al_2_O_3_ nanofibres, developed in [35], have an extremely high aspect ratio of about 10^7^ and a controllable diameter from 5 to 50 nm. Such an approach not only prevents Gr agglomeration during sintering, but also allows to control of the electron conductivity percolation threshold due to the small amount of graphene on the surface of nanofibres and its controlled orientation in the composite. As it was shown in [35], the incorporation of GAlN into the Al_2_O_3_ matrix can improve the fracture toughness by about 60% with no significant decrease in hardness. Moreover, the high electrical conductivity of alumina-based composite (1.51 S/cm^−1^) has been archived upon 15 wt.% addition of GAlN to alumina matrix. The properties of GAlN doped composites with tetragonal zirconia matrix were investigated in [36]. It was shown that the percolation threshold was achieved at 5 wt.% GAlN content, which is equal to ~0.5 wt.% Gr. The introduction of GAlN to cubic zirconia matrix seems to be prospective to achieve balanced ionic and electronic conductivity. From the literature analysis, it is seen that the percolation threshold highly depends on the derivative type and the ceramic matrix (cubic, tetragonal zirconia) it varies from 1 to 5 wt.% of graphene additive. So, the optimal amount of GAlN should be established experimentally or via modelling in each case. Thus, the goal of the present work was to investigate the effect of GAlN on the phase composition, conductivity and mechanical properties of the YSZ ceramics matrix produced by spark plasma sintering.

## 2. Experimental

### 2.1. Synthesis of YSZ/GAlN Composite Ceramics

YSZ nanosized powder synthesis. Nanosized powder of zirconia stabilized with 9 mol.% Y_2_O_3_ was synthesized by sol-gel method from aqueous salts solution in a variant of the reversed co-precipitation technique according to [9,37]. Starting salts of Y(NO_3_)_3_·6H_2_O (Acros Organics, Geel, Belgium, 99.9%) and ZrO(NO_3_)_2_·6H_2_O (Acros Organics, Geel, Belgium, 99.5%) were taken in a certain molar ratio and dissolved in distilled water to prepare a solution having a total concentration of 0.1 M. Precipitation was realized by dropwise addition of salts solution to 1 M ammonia solution with a speed of 1–2 mL/min (LenReactiv Ltd., St Petersburg, Russia, c.p.) keeping the temperature of the synthesis medium 274–275 K via ice bath. The obtained precipitate was aged for 24 h, when washed until neutral pH was reached to form a gel. The latter was freeze-dried to obtain nanosized powder (Labconco, 1 L chamber, Kansas City, MO, USA; 293 K, 24 h, *P* 0.018 mm Hg). The YSZ powder was when milled (Pulverisette 6, Fritsch, Germany, 420 rpm, 24 reverse cycles of 5 min each) in order to further diminish powder agglomerates size, when annealed at 1073 K for 3 h to allow the formation of cubic solid solution and when milled again with the same conditions for mechanical activation. Thus, YSZ nanosized powder precursor was obtained. The XRD pattern of the obtained amorphous powder is presented in Figure 1.

GAlN/YSZ composites manufacturing. Graphene-augmented γ-Al_2_O_3_ nanofibres (GAlN) were used as a modifier for ceramic composite manufacturing. For that, GAlN were successfully synthesized in the group of prof. Hussainova according to techniques listed in [28,30,32] and provided to authors for the present experiment. The γ-Al_2_O_3_ nanofibres were synthesized by controlled aluminium liquid phase oxidation The nanofibres were subjected to the chemical vapour deposition (CVD) process at 1273 K using N_2_ and CH_4_ gas mixture, and were cooled to room temperature in a stream of pure N_2_ [38]. The resulting GAlN consisted of the uniformly cover ANF with a diameter of 7–10 nm covered by graphene layers with a thickness of 0.3 nm, which ensures its strong bonding of graphene with alumina [34]. The individual components (YSZ nanopowder and GAlN) were mixed in different ratios. The GAlN addition was varied from 0 to 5 wt.% (see, Table 1), being equal to 0–0.5 wt.% of graphene addition. The compositions of GAlN/YSZ composites were chosen based on the literature data on the optimal amount of graphene additive to achieve the percolation threshold in graphene-zirconia and GAlN-zirconia composites [10,18,19,20,36]. The composite powders were placed in a graphite die with a 20 mm diameter and consolidated by spark plasma sintering technique (FCT Systeme GmbH, Germany) at 1673 K with applied uniaxial pressure of 75 MPa in an N_2_ atmosphere for 15 min. Sintered specimens were then cooled to room temperature (with an approximate speed of 30 K/min). The conditions of sintering were chosen based on data from [28,30,32]. The temperature measurement took place in the vicinity of the powder compact centre, which gives a much more correct value than the measurement of the die temperature.

### 2.2. GAlN/YSZ Composites Characterization

The phase compositions of the GAlN/YSZ composites were examined via X-Ray Diffraction technique (XRD, Shimadzu XRD 6000, Cu-K_α_ irradiation with wavelength 1.5406 Å). Phase identification was carried out using the Powder Diffraction File database (PDF-2, 2021 [39]). Crystallinity was calculated using the standard software package of a diffractometer. Crystallite sizes were estimated using the Scherrer equation:(1)$$d_{cryst}=\frac{K\lambda }{\beta \cos\theta }$$
where *d_crys_*_t_—is the crystallite size in nm, *K*—Scherrer constant (~0.9 for spherical particles), *λ*—X-ray wavelength, *β*—peak full width at half maximum (FWHM) corrected to FWHM of the internal standard, and *θ*—Bragg angle. The peak at 2Θ = 29.5° was taken for the estimation for each ceramic composite. States of GAlN and YSZ matrix in the composites were investigated via Raman spectroscopy (Raman spectrometer Horiba, LabRam HR800). The measurement conditions were as follows: backscattering mode, He-Ne laser with wavelength 633 nm and powder of 0.08 mW, 50× lens, spectrum gathering time 4 × 300 s, diffraction grating 600 lines/mm. Microstructures of sintered composite ceramics, fracture surfaces and ceramics after thermal cycling experiments were studied via high-resolution scanning electron microscopy (high-resolution scanning electron microscope Zeiss Merlin, accelerating voltage 20 kV). Energy-dispersive spectroscopy (EDS) and element mapping were carried out using the equipment of the Zeiss Merlin electron microscope, X-ray microanalyzer Oxford Instruments INCAx-act. The microhardness of sintered composites was measured via the Vickers hardness method (Tester Shimadzu HMV-G21DT with CCD camera microscope). Testing was performed with the applied force of 2.9 N (HV0.3). The data was averaged on 15 measurements for each composite specimen. The surfaces of the samples were prepared by the same polishing procedure as for microstructure investigations.

### 2.3. Electrochemical Testing of GAlN/YSZ

The electrochemical behaviour of GAlN/YSZ composites was investigated via electrochemical impedance spectroscopy (Autolab PGSTAT 302 N Potentiostat/Galvanostat). Rectangular central sectors with linear dimensions of 10 × 10 × 5 mm were cut for the measurements. Silver electrodes in a form of a paste then were braised on the surfaces of the composite samples. Prepared samples were then dried in a muffle chamber at 333 K for 2 h for evaporation of the residual paste solvent. The electrochemical response of the composites was measured in 100 Hz–1 MHz frequency range using cyclic heating-cooling from 473 to 1073 K with 50 K steps in the argon atmosphere (national standard of Russia, GOST 10157-2016, equal to ISO 2435-73). The residual partial pressure of oxygen in the atmosphere was no more than 10^−5^ atm. Measured impedance spectra were treated using «Impfco» software, as a package of the Potentiostat (Metrohm, Switzerland). Activation energies of conductivity were estimated using the Arrhenius equation from linear fits regressions of the temperature dependence of conductivity (2):(2)$$\sigma =\sigma_{0}\exp\left(\frac{E_{a}}{RT}\right)$$
where $E_{a}$—activation energy, R—universal gas constant, T—absolute temperature, $\sigma_{0}$—pre-exponential factor.

## 3. Results

### 3.1. The Structure of Sintered Composites

XRD patterns for GAlN/YSZ ceramic composites are presented in Figure 2. The XRD patterns obtained for all samples correspond to a single phase, i.e., cubic zirconia-based solid solution without the admixtures of low symmetric tetragonal or monoclinic zirconia phases. The peaks corresponding to the presence of the GAlN phase (carbon or γ-Al_2_O_3_) are absent even for compositions with 3 and 5 wt.% of the graphene-containing additive. Remarkable that the ZrC phase is present in the XRD patterns for 0 GAlN, 1 GAlN and 5 GAlN, which indicates that the reaction between the composite powder and the graphite die took place during sintering.

A comparison of XRD patterns for 0 GAlN and 5 GAlN composites after synthesis and polishing with a diamond suspension is present in Figure 3. Polishing results in ZrC removal from the surface and characteristic peaks for the ZrC phase at 2Θ = 34, 38, 55, 66 and 69° are absent in the following XRD patterns. So, one can see that the reaction between YSZ powder and graphite takes place only on the thin surface layer of the composite during SPS.

Composites SPS-ed under chosen conditions is fully dense with relative density values of 98.7–99.4% (see Table 2). The introduction of GAlN has almost no effect on the density of composites. A different picture is observed when it comes to crystallinity (the ratio of crystalline to amorphous phase) of the samples. Nanosized zirconia-based precursor after annealing at 1073 K has a crystallinity value of ~30%, which is typical for oxide powders obtained via the sol-gel coprecipitation technique [40]. A broadening of the peaks is observed in XRD patterns of composites indicating that the crystalline structure is not formed completely. Indeed, the crystallinity value of the SPS-ed 0 GAlN reference sample is 66%, which is in good correlation with short dwelling time. The addition of 0.5–5 wt.% GAlN causes a close-to-linear increase of crystallinity up to 84% accompanied by the peaks narrowing in the corresponding XRD patterns. The sizes of crystallites estimated according to Equation (1) for single-phased polished samples change in accordance with crystallinity degree upon the increase of GAlN content and vary from 87 to 104 nm

For further investigation of the carbon state and YSZ after sintering, Raman spectroscopy was used. The spectra obtained for the cross-section of the samples are presented in Figure 4. Each spectrum obtained for composites can be divided into two regions: (1) the region from 0 to 1000 cm^−1^ corresponding to signals from the ceramic YSZ phase and (2) the region from 1300 cm^−1^ to 3000 cm^−1^ corresponding to signals from the carbon phase (Figure 4a). All the peaks in the spectrum (160, 260, 610, 680 cm^−1^) correspond to the cubic stabilized zirconia phase (Figure 4b) [41,42]. The signals from the ceramic phase decrease with the increase of GAlN content. For cubic ZrO_2_ the number of fundamental vibrations is 33. In fact, due to the high symmetry of the structure, the vibrations deteriorate and typically 3 or 4 bands appear in the Raman spectrum of cubic ZrO_2_ A band about 610 cm^−1^ is a main band (zone-centre F_2_g mode) corresponding to the vibration of the O-Zr-O atoms. The band at 680 cm^−1^ is related to the disorder-induced scattering [42]. The decrease of the intensity of the band at 680 cm^−1^ in spectra obtained for 1 GAlN–5 GAlN composites indicates the structural ordering of the composites and is in accordance with the crystallinity and crystallite sizes increase in the composites (see Table 2). In turn, the intensities of peaks, related to graphene, at 1330, 1590 and 2660 cm^−1^ increase [43,44]. The calculated I_D_/I_G_ and I_2D_/I_G_ intensity ratios are presented in Table 2. From I_D_/I_G_ < 1 one can see that the structure of graphene was well preserved in the structure. It should be noted that the 2D peak corresponding to the in-plane breathing-like mode of the carbon rings, is present in spectra for composites 1 GAlN–5 GAlN. The ratios of 2D and G peaks intensities being less than one, indicate high structural integrity and low degree of damage of graphene on the surface of alumina fibres after sintering. Typical spectra collected from the surface of 0 GAlN and 5 GAlN composites after polishing are shown in Appendix A.

The microstructures of the composites reflect the discussed features of the phase composition of the composites. Microphotographs of cross-sections of the sintered composites are presented in Figure 5. Indeed, two zones can be distinguished in the cross-section of each composite: an edge layer and a bulk layer. The presence of two zones becomes more pronounced with the increase of GAlN content, i.e., in the case of 0 GAlN, 0.5 GAlN and 1 GAlN edge layer are slightly visible, whereas in the photos obtained for 3 GAlN and 5 GAlN, they are clearly observed. Moreover, the thickness of the formed edge layer increases from ~450 µm for 0 GAlN to ~600 µm for 3 GAlN, and ~750 µm for 5 GAlN. Edge layer is more uniform and possesses defects compared to the bulk of specimens.

The detailed SEM images obtained for 0 GAlN ceramics are presented in Figure 6. Both edge and bulk layers represent a microstructure, being typical for YSZ ceramics. However, the bulk is more uniform almost with no pores (Figure 6b), whereas the edge layer has more defects in the structure (Figure 6a,c). Rather coarse grains micron-sized grains were formed during SPS-sintering (see, Figure 6c,d).

Some changes in microstructure are observed upon the SPS of composite powder with 0.5 wt.% GAlN content (Figure 7). The addition of GAlN results in the formation of almost non-porous ceramic matrix grains with micron-sized grains. The grains with angles close to 90° separated by rather thin grain boundaries are seen in the edge layer. When it comes to the bulk, the grains are less pronounced. The small pores along the grain boundaries allow us to estimate their size to be equal to ones of the edge layer. Elemental mapping of Al was performed for Figure 7c,d to estimate GAlN distribution in structure (Figure 7e,f). As seen from Figure 7e,f aluminum is distributed almost homogeneously in the bulk of the composite, indicating the presence of GAlN in the structure of the composite. Small amounts of aluminum (and thus GAlN) are segregated in the pores of the composite. Edge layer is enriched by aluminum. The segregation of GAlN along the grain boundaries is clearly observed.

Significant microstructure changes are observed in composites with higher GAlN content. SEM images for 3 GAlN are presented in Figure 8. Grain refinement and the elongation of grains take place up to 3 wt.% GAlN introduction to the ceramic matrix. Additionally, bright inclusions of crystals with cubic shapes are seen in the edge layer of the composite (Figure 8b). GAlN is present in the structure of the edge layer. They correspond to ZrC, having face-centered cubic diamond structure, space group $Fm\bar{3}m$. Composite bulk does not have inclusions of zirconium carbide crystals. The data obtained is in accordance with XRD results. In order to estimate GAlN distribution in the structure of 3 GAlN composite, the EDS mapping has been performed along the cross-section of the composite sample (see, Figure 8d,e). The results, presented in Figure 8f, show that zirconium and oxygen are homogeneously distributed in the grains. Whereas, in pores, the Zr element is almost absent and the content of oxygen is higher than its content in grains. The high content of aluminum is present in the pores. Small amounts of Al are seen also along the grain boundaries. So, one can conclude that graphene-augmented Al_2_O_3_ nanofibres are located along grain boundaries and slightly segregate in the structural defects.

The microstructure of the5 GAlN composite is presented in Figure 9. In comparison with composites having smaller content of GAlN additive, three zones can be distinguished in the structure of the composite: edge layer, the intermediate layer and the bulk. The edge layer contains a significant amount of ZrC crystals grown in the pores and other defects (Figure 9a,b). Indeed, according to EDS mapping shown in Figure 10d, the crystals are enriched with zirconium (Figure 10c) and depleted with oxygen (Figure 10b). All structural defects of the edge layer are also enriched with aluminum, indicating the presence of GAlN even in the edge layer of the composite. The intermediate layer is characterized by the presence of well-distinguished elongated micron-sized grains of different sizes (Figure 9c,d). Small amounts of zirconium carbide are present along the grain boundaries and in the defects of the individual grains. Bulk has a structure close to the intermediate layer, but is refined. It consists of elongated grains, separated by well-distinguished thin grain boundaries.

An interesting phenomenon has been observed during elemental mapping (Figure 10). As for previous composites, the defects in the bulk layer are enriched with aluminum and oxygen (Figure 10f,h) and depleted with zirconium (Figure 10g), which points to the presence of GAlN along the grain boundaries.

The results of microhardness tests are presented in Figure 11. Testing was performed for both for overall composite surface and for different phases of polished composites 1 GAlN–5 GAlN. The ceramics sample 0 GALN and 0.5 GAlN show superior Vickers hardness values of 15.6 ± 2.4 and 15.8 ± 1.5 GPa, respectively. A small addition of 0.5 wt.% GAlN does not affect the mechanical properties of the composite material but results in a smaller experimental error, i.e., better-formed structure of composite. When the addition of GAlN reaches 1 wt.%, the number of ZrC crystals (white inclusions) becomes sufficient to perform hardness testing of the phase. Indeed, hardness for the areas is much higher compared to the YSZ matrix: 14.9 ± 1.1 GPa vs. 11.3 ± 1 GPa. The value of ~11 GPa is typical for fully stabilized zirconia ceramics [45].

Upon 3 wt.% GAlN addition hardness increase is observed. The difference in hardness between the ceramic matrix and ZrC becomes almost equal within the experimental error, being 15.8 ± 0.7 GPa and 15.1 ± 1.4 GPa. When it comes to the 5 GAlN composite the hardness of the composite decreases to 12.6 ± 1.6 GPa. The test of the separate sites of the composite corresponding to ZrC polycrystals agglomerate shows an exceptional hardness of 17.9 ± 1.3 GPa. Along with that, the hardness of individual ZrC crystals in ceramic matrix increases to 16.7 ± 1 GPa.

### 3.2. Electrochemical Studies of the Composite

The results of the electrochemical studies obtained for the 0 GAlN ceramics and 0.5 GAlN composite are presented in Figure 12. Each impedance spectrum of 0 GAlN consists of two arcs referred to as the polycrystalline solid ionic conductor [46]. However standard temperature step of 50 K does not allow us to fully reveal both arches. Indeed, at 475 K the arch at high frequencies (low Z_real_ values area) is clearly seen. It refers to the resistivity of grains of polycrystalline ceramics. Only the beginning of the arch in the area of the low frequencies is seen at the temperature. It refers to the resistivity of grain boundaries (Figure 12a). The addition of 0.5 wt.% of GAlN to the ceramic matrix does not change the ionic character of the conductivity (Figure 12c). The impedance spectra obtained for 0.5 GAlN composite is similar to ones for 1 GAlN and has one clear arch at high frequency (Figure 13a) corresponding to the ionic contribution of grains and grain boundaries to the total conductivity. Both arches can be seen in the spectra obtained for 0.5 GAlN at temperatures higher than 600 K (see Figure 13b). Due to frequency range limitations, two arcs may overlap with each other or are not be seen at lower temperatures.

The values of the total conductivity were calculated in all measured temperature ranges from the impedance data. The temperature dependencies of conductivity obtained for SPS-ed composites are presented in Figure 13a. The data for 0 GAlN ceramics is close to YSZ ceramics sintered using equilibrium techniques (annealing in the air) and reaches 5×10^−5^ S/cm at 673 K with the lower activation energy of conductivity of 0.99 eV (see Table 3 and Figure 13) [47]. Upon the temperature increase, the conductivity trend deviates from linear Arrhenius dependence and conductivity reaches 10^−4^ S/cm at 773 K. For that reason, the measurements have been repeated and resulted impedance plots are shown in Figure 12b. After the second heating-cooling cycle the overall conductivity decreases for about a half order of magnitude. The linear dependence is seen in the whole temperature range with the activation energy of 1.01 eV (see Figure 13a and Table 3). The maximum conductivity value obtained for 0.5 GAlN composite reaches ~3·10^−3^ S/cm at 923 K. Temperature dependence of conductivity obtained for the composite also fits the linear Arrhenius dependence (see Figure 13a). For the composite values of grain boundaries conductivity is higher than the ones of grains. Along with that, the activation energies of conductivity were calculated for both components: grains and grain boundaries. The obtained values are 0.77 and 0.88 eV, respectively being lower than observed for YSZ ceramics (see Table 3).

At high temperatures, >923 K, the conductivity values turn out to be unstable; a gradual transition of the spectrum to a higher Z_re_ area takes place, which leads to the respective decrease of conductivity. The exposure of the composite for 30 min at 973 K results in the conductivity of the grain component change in one order of magnitude, from 3·10^−4^ S/cm to 5·10^−5^ S/cm. This process is accompanied by the gradual destruction of 0.5 GAlN composite into the parts. The other samples: ceramics and composites were stable after the experiments and did not crack into pieces. The process of 0.5 GAlN composite destruction is probably caused by the thermal stresses developed in the composite upon the small addition of GAlN.

The electrical behaviour of 1 GAlN composite is similar to 0.5 GAlN one (Figure 14). The difference is that the shapes of the arcs in the impedance spectra deviate from the ideal semicircle in all investigated temperature ranges (Figure 14a). During the first cycle of heating the conductivity value reaches ~10^−3^ S/cm at 773 K. The linear dependence of conductivity is observed with the reverse temperature at all the investigated range with rather a low activation energy of conductivity, 0.74 eV (see Table 3, Figure 13a). Similarly, to 0.5 GAlN composite, a long exposure of 1 GAlN composite at high temperatures results in the gradual loss of conductivity (Figure 14b). The conductivity changes from ~10^−3^ S/cm to 3·10^−4^ S/cm after exposure for 90 min at 773 K. During the second measurement, the conductivity change is accompanied by the change of activation energy of conductivity to 1.11 eV (see Table 3). Thus, the sample begins demonstrating a behaviour typical for YSZ ionic conductors (see Figure 13a) [46]. The results of impedance measurement obtained for 3 GAlN are shown in Figure 14c,d.

In a temperature range from room to 773 K, the percolation level is reached: 3 GAlN composite possesses high electronic conductivity without polarization arc or arcs referred to grain or grain boundaries ionic conductivity (Figure 14c). At room temperature, the conductivity value is ~10^−2^ S/cm. The linear increase with the reverse temperature is observed and conductivity reaches 4·10^−2^ S/cm at 773 K. The activation energy calculated from the slope is 0.06 eV (Figure 13a). At temperatures higher than 822 K, the obtained impedance spectra change drastically: the ohmic resistance increases and an arc referred to as polarization appears. At 871 K, the ohmic resistance of the composite slightly increases, whereas polarization resistance decreases. Further heating causes the gradual increase for both ohmic and polarization components of conductivity with the maximum values of 9·10^−3^ S/cm and 6·10^−3^ S/cm, respectively. The calculated activation energies for the ohmic and polarization components are almost equal being 0.2 eV and 0.26 eV, respectively (see Table 3). Upon the second heating cycle, the shapes of the impedance spectra obtained changed significantly (Figure 14d). Each spectrum obtained in the temperature range of 298–772 K consists of one arch at high frequency. The second arch is slightly observed at 523 K (green spectrum in Figure 14d). It is typical for the polycrystalline ionic conductor. The total resistance changes by six orders of magnitude during the heating to about 200 K. The conductivity linearly increases up to 2.7·10^−4^ S/cm at 723 K with the reverse temperature. Because of frequency limitations, only arc referred to grain conductivity can be seen. Along with that, the calculated activation energy of conductivity reaches 1.1 eV after the second measurement. So, a change from electronic to ionic behaviour is reached. The 5 GAlN composite also demonstrates the electronic behaviour (see Figure 15 and Table 4). At the temperatures from room to 771 K the values of conductivity lie in a range from 0.12 to 0.14 S/cm. The obtained values of the conductivity do not fit the linear Arrhenius dependence. Once 798 K is reached, the resistivity starts increasing and the polarization arc appears. When cooled down to room temperature, the composite does not possess the electronic component of conductivity anymore and the total conductivity level drops to 2·10^−8^ S/cm.

In order to obtain more detailed data on composite stability, the thermal evolution of conductivity was separately investigated for temperatures of 771 K and 784 K, the obtained impedance spectra are demonstrated in Figure 15b and in Table 4. After exposure at 771 K for at least 40 min the conductivity level decreases non-significantly from 0.127 S/cm to 0.07 S/cm. The shape of spectra remains unchanged: only ohmic resistance is seen with no arcs typical for or ionic conductivity or any polarization phenomena. In contrast, during the measurement at 784 K noticeable changes in the shape of the spectrum occur even after 15 min of exposure (Figure 15a). After 30 min of exposure at 784 K, the polarization arc predominates in the spectrum. After 85 min of exposure, the ohmic resistance drops to 20 Ohms and polarization resistance—to ~160 Ohms.

## 4. Discussion

Previously, it was reported that GAlN efficiently enhances the sinterability of the Al_2_O_3_ ceramic matrix during the SPS process [5,34]. When it comes to the zirconia matrix, similar phenomena are observed. Indeed, the crystallinity of composites increases almost linearly from 66 to 84% upon the increase of GAlN content to 5 wt.%. The crystallite sizes, estimated using Scherrer’s formula (Equation (1)) show a similar trend indicating a better formed crystal structure. Raman spectroscopy data also indicated the structural ordering in the composites up on GAlN content increase (see, the decrease of the intensity of the band at 680 cm^−1^ in the spectra obtained for composites in Figure 4). The fractography results obtained for ceramics and composites differ significantly (see, Figure 16). The SEM photo of fracture surfaces, obtained for 0 GAlN ceramics generally demonstrates a fracture typical for a polycrystalline solid. At the same time, the characteristic features of amorphous phase fracture are clearly observed (indicated by the orange circles in Figure 16a), being in accordance with the relatively low crystallinity of 0 GAlN (66%). Figure 16b–d demonstrates the fracture of the composites. The fracture also takes place along the grain boundaries of the grains in 1 GAlN composite. The features of the “glassy-type” cracking typical for amorphous solids are also seen in the composite. Under higher magnification, one can see that GAlN is pulled out from the fracture surface leaving the cavities in the structure.

From the data obtained it is seen that GAlN addition enhances the crystallization resulting in the structural ordering of the composites compared to ceramics. It can be assumed that GAlN significantly affects the mechanism of zirconia matrix sintering. According to [32], the addition of graphene to the zirconia matrix results in the mixed mechanism of sintering with a grain boundary diffusion predominance. Indeed, the increase of graphenated alumina nanofibres with a super-high aspect ratio (the length-to-diameter ratio is ∼100) to 3–5 wt.% induces the overall elongation of grains and formation of the grain boundaries network in zirconia-based composites (see Figure 5, Figure 6 and Figure 7). The introduction of 0.5 wt.% resulted in well-distinguished grains and grain boundaries as well as the possibility to separate impacts of grains and grain boundaries into the ionic conductivity from impedance spectroscopy data (see, Appendix A). The increase of crystallinity and structure refinement results in about an order of magnitude increase in conductivity of 0.5 GAlN and 1 GAlN composites compared to SPS-ed ceramics (see Figure 13a). The presence of a large amount of the amorphous phase plays a negative role in the conductivity. The more accomplished a crystal structure of a solid is, fewer defects are present in the structure, the higher the conductivity [48,49,50].

Both zirconia-based ceramics and composites have pronounced heterogeneity in the near-edge layer with a length of 450–750 µm. The edge layer having a different microstructure than the bulk is present in 0 GAlN ceramics with no graphenated alumina nanofibres added. The obtained microstructures are likely a feature of the SPS technique. The microstructure data obtained is in accordance with the recent results obtained by authors during SPS sintering of reduced graphene oxide doped zirconia composites [9]. The technique belongs to a group of non-equilibrium techniques. Simultaneous application of current and pressure during SPS results in a non-linear increase of the material’s conductivity and its rapid heating. Since the conditions are non-equilibrium, the amount of current passing through the near-surface layer and the bulk of the sample can differ significantly. The edge layers of all samples obtained have a loose microstructure and more defects compared to a bulk. Besides, the conditions of sintering induce a concentration gradient of carbon from the graphite due to the sample bulk. As a result, a side reaction of ZrO_2_ reduction to ZrC is observed on the surface of ceramics and composites studied. Despite the added amount of GAlN (0.5–5 wt.%) being equal to 0.05–0.5 wt.% graphene, the increase in GAlN content induces the increase of the microstructure heterogeneity. As seen from EDS and Vickers testing results (see, Figure 10 and Figure 11) the addition of GAlN results in small amounts of zirconium carbide formed in the composites bulk. At high concentrations, GAlN also may react with the zirconia matrix. It results in the superior hardness of 0 GAlN–0.5 GAlN and 3 GAlN of 15.1–15.8 GPa. The obtained results are about 15% higher than reported for cubic YSZ ceramics and are close to the data reported for the tetragonal phase of 3 mol.% yttria zirconia (3YTZP) ceramics [51].

The electrical properties of the SPS-ed GAlN composites are of particular interest. The 0.5 GAlN and 1 GAlN composites are ionic conductors in all investigated temperature ranges. The Arrhenius dependences of conductivity on reverse temperature are observed for the composites (see, Figure 13). The activation energies of conductivity, calculated from the slopes of Arrhenius dependences of conductivity for 0.5 GAlN and 1 GAlN are considerably lower than values for YSZ ceramics (0.78, 0.74 eV vs. 1.1 eV, see Table 3) [50] and similar the ones obtained for SPS-ed YSZ/rGO composites [9]. Rather small additions of GAlN result in the microstructure refinement and crystal accomplishment (well-formed grains and grain boundaries) and thus allow to decrease the potential barrier for a charge carrier to overcome for a charge transfer both though the bulk of composite and along the grains boundaries. The additional cycle of heating results in the activation energy increase to 1.11 eV with no loss in conductivity at high temperatures which is most probably due to graphene destruction during heating (see Table 3 and Figure 13). The assumption is confirmed by STA data (see, Appendix A). The percolation threshold is reached upon the addition of 3 wt.% GAlN in 3 GAlN composite (see Figure 14) and high electronic conductivity of composite appears. The amount of GAlN added for the percolation threshold is consistent with the data obtained by Belmonte et al. for graphene/YSZ composites (7.1 vol.%) [10] and recent work of authors for rGO/YSZ composites [9]. Electric conductivity dominates at temperatures from 298 to about 800 K, provided by a three-dimensional network of GAlN formed in the ceramic matrix. At higher temperatures, a mixed conductivity is observed. Remarkable, that the difference between the conductivity values is less than one order of magnitude. One can suggest that due to the low amount of pristine graphene in GAlN (0.3 wt.%) does not block the connection between the grains and allow fast charge transfer through the bulk of the ceramic matrix. Above the threshold, the electronic component dominates over the ionic one. Here the diffusion of oxygen charge carriers seems to be blocked by GAlN present along the grain boundaries of the ceramic matrix. Thus, the ionic component of conductivity is hindered in all investigated temperature ranges.

The stability and thermal evolution of conductivity in the SPS-ed ceramics and composites were investigated in time and in the temperature range of 298–973 K in the argon atmosphere. As seen from Figure 13b, the changes in the shape of the arcs in impedance spectra are observed even for 0 GAlN ceramics. Most likely, it is due to the fact that, the phase equilibrium was not reached during the fast SPS process. The hypothesis is confirmed by the low crystallinity of as-sintered 0 GAlN ceramics (66%, see Table 2). At small amounts of GAlN the ionic conductivity of the grains in 0.5 GAlN and 1 GAlN decreases by about one order of magnitude. Similar processes take place in 3 GAlN composite. Only ionic conductivity is observed after the second measurement cycle. At the same time, the conductivity of the 5 GAlN composite is stable in time at 771 K. The increase of temperature to 784 K results in a steady conductivity drop of one order of magnitude. Prolonged exposure result in further conductivity decrease. The latter is followed by the crystallinity increase from 87 to 92% according to XRD data. The obtained data is in accordance with the model of conductivity evolution with temperature in rGO/YSZ composites, which has been suggested recently by authors in [9]: the thermal cycling results in the grain growth and ceramic network expansion. The microstructures of 0.5 GAlN and 3 GAlN composites after thermal cycling are shown in Figure 17.

As seen, the bulk of 0.5 GAlN consists of slightly coarser grains comparing to the one after sintering. The overall structure remains the same. At the same time structure of 3 GAlN composites after thermal cycling changes considerably. The evolution of grains from elongated to close-to-cubic shapes is observed. Latter results in the distortion and partial breakup of the GAlN network. Besides, the composites were studied by STA. The exothermal peak almost not accompanied by the mass loss is observed in the DSC curves of all composites (see Appendix A). Since the amount of pristine graphene in 3 GAlN and 5 GAlN composites is only 0.3–0.5 wt.%, the peak in the DSC curve confirms partial graphene degradation and its burnout from the surface of alumina nanofibers. Thus, even short heating promotes the removal of a significant amount of graphene and subsequent conductivity change. At the same time at 5 wt.% GAlN addition the conductivity value remains stable up to 770 K.

In summary, novel GAlN composites enhanced with 0.5–5 wt.% graphene-augmented alumina nanofibers (equal to graphene amount 0.005–0.5 wt.%) were obtained via SPS using stabilized zirconia nanopowder. The percolation threshold and mixed ionic-electronic conductivity were obtained for the 3 GAlN composite. The obtained difference between electronic and ionic components is a half magnitude and even less, and superior Vickers hardness close to the data for tetragonal zirconia make the composite attractive as a material for solid oxide fuel cells and oxygen pumps. Without a doubt, further research in the field is necessary to improve the thermal stability of the composites.

## 5. Conclusions

Via SEM, XRD and Raman spectroscopy it was shown that the use of the graphene-augmented alumina nanofibres (GAlN) allows producing fully dense composites with an accomplished crystal structure and elongated grains. Vickers hardness testing showed excellent hardness, >15 GPa, due to the small amount of ZrC formed. Via SEM, EDS and impedance spectroscopy it was shown, that 0.5–1 wt.% GAlN addition results in the increase of conductivity in one order of magnitude in the whole investigated temperature range followed by the decrease of the activation energies of conductivity from 0.99 to 0.74 eV compared to SPS-ed ceramics. Via impedance spectroscopy, it was shown that the percolation threshold is observed upon 3 wt.% GAlN addition with the activation energies of conductivity 0.02 eV. 3 GAlN composite demonstrates an electronic conductivity at low and intermediate temperatures and mixed ionic-electronic conductivity at elevated temperatures. Via SEM it was shown that the thermal cycling results in the grain evolution from elongated to cubic grains and GAlN network. The electronic conductivity level of the 5 GAlN composite is stable in time at 771 K.

## Figures and Tables

**Figure 1 materials-16-00618-f001:**
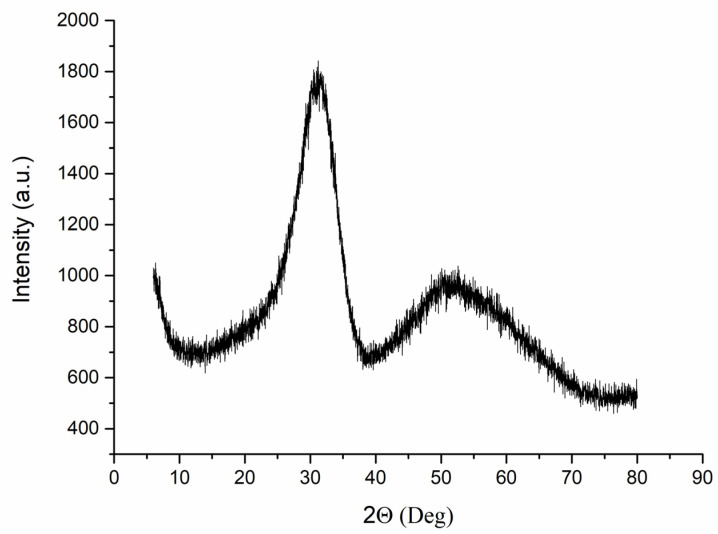
XRD pattern of the amorphous YSZ nanopowder after synthesis.

**Figure 2 materials-16-00618-f002:**
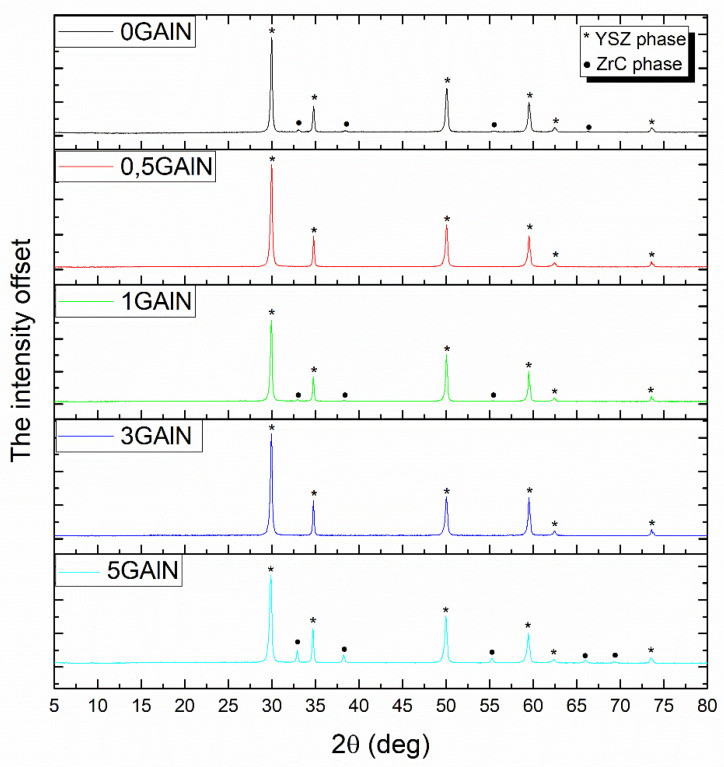
XRD patterns of obtained YSZ-GAlN composites.

**Figure 3 materials-16-00618-f003:**
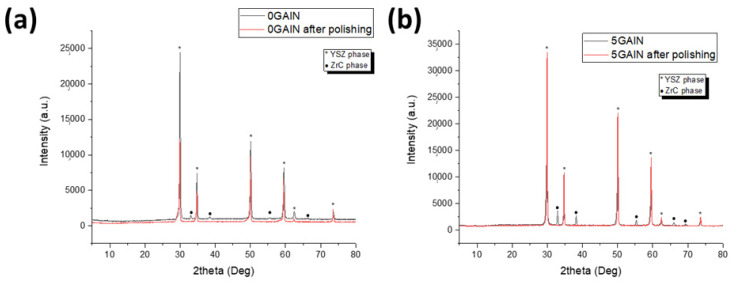
XRD patterns for (**a**) 0 GAlN and (**b**) 5 GAlN composites before and after polishing.

**Figure 4 materials-16-00618-f004:**
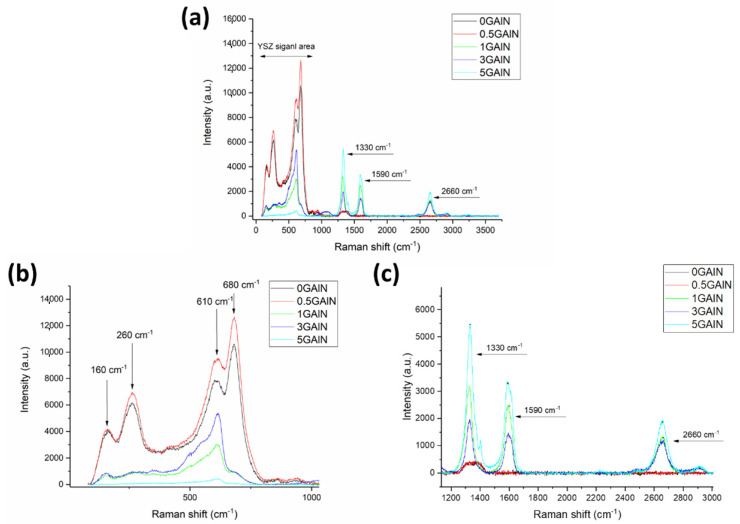
Raman spectra of the manufactured YSZ/GAlN composites: (**a**) the whole spectra, (**b**) zoomed area from 0 to 1000 cm^−1^ (**c**) zoomed area from 1200 to 3000 cm^−1^.

**Figure 5 materials-16-00618-f005:**
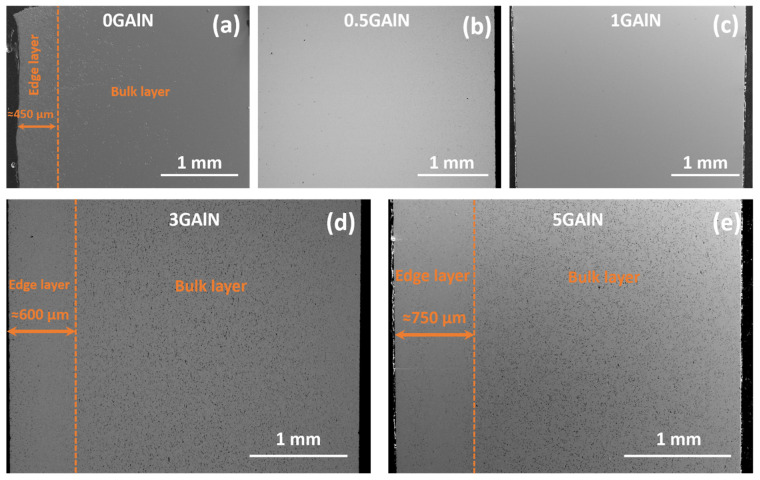
Scanning electron microscopy images of the sintered YSZ/GAlN composite ceramics: (**a**) 0 GAlN, (**b**) 0.5 GAlN, (**c**) 1 GAlN, (**d**) 3 GAlN, (**e**) 5 GAlN. Magnification 70×.

**Figure 6 materials-16-00618-f006:**
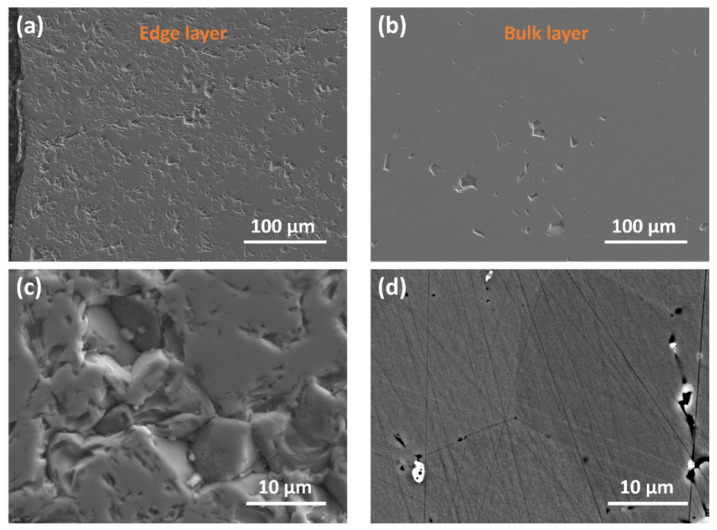
SEM images obtained for 0 GAlN ceramics: (**a**,**c**)—edge layer (**b**,**d**)—bulk layer.

**Figure 7 materials-16-00618-f007:**
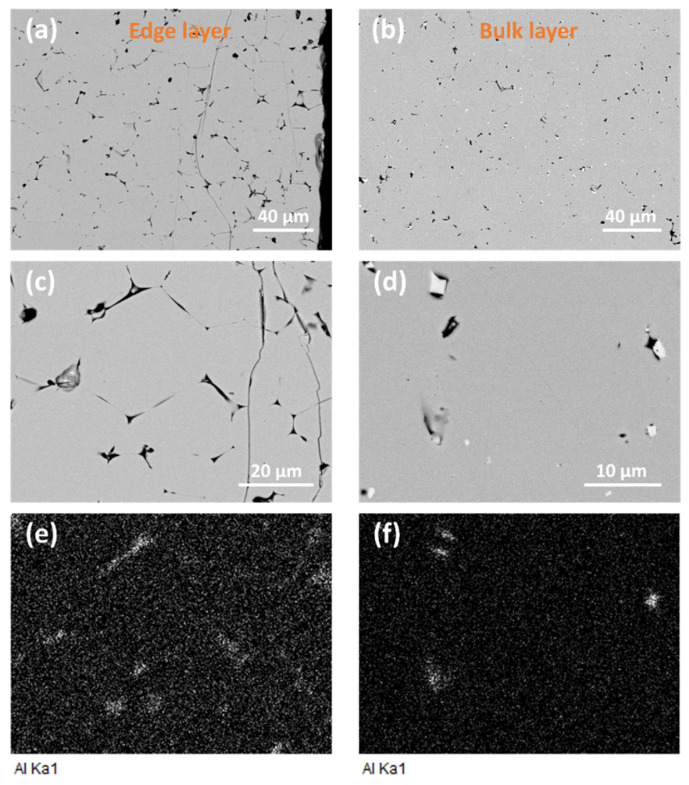
(**a**–**d**) Microstructure of 0.5 GAlN composite ceramics, (**e**,**f**) aluminum elemental maps, taken from images (**c**) and (**d**), respectively.

**Figure 8 materials-16-00618-f008:**
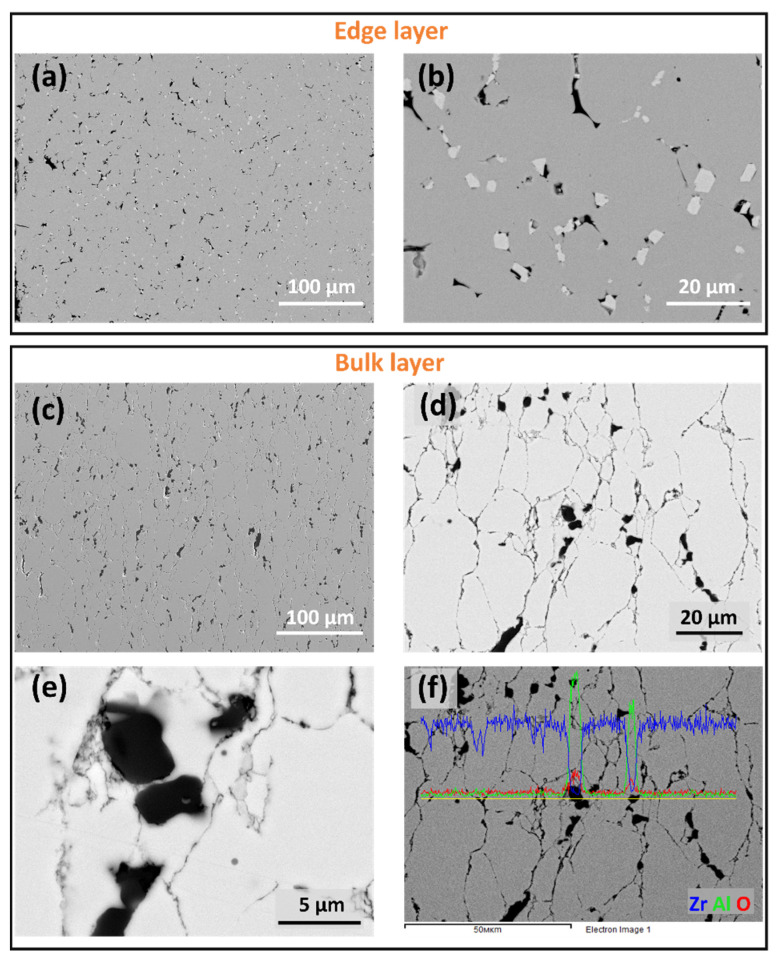
SEM images for 3 GAlN composite: (**a**,**b**)—edge layer, (**c**–**e**) bulk layer; (**f**) EDS signal for Zr, Al, O elements, measured along the sample.

**Figure 9 materials-16-00618-f009:**
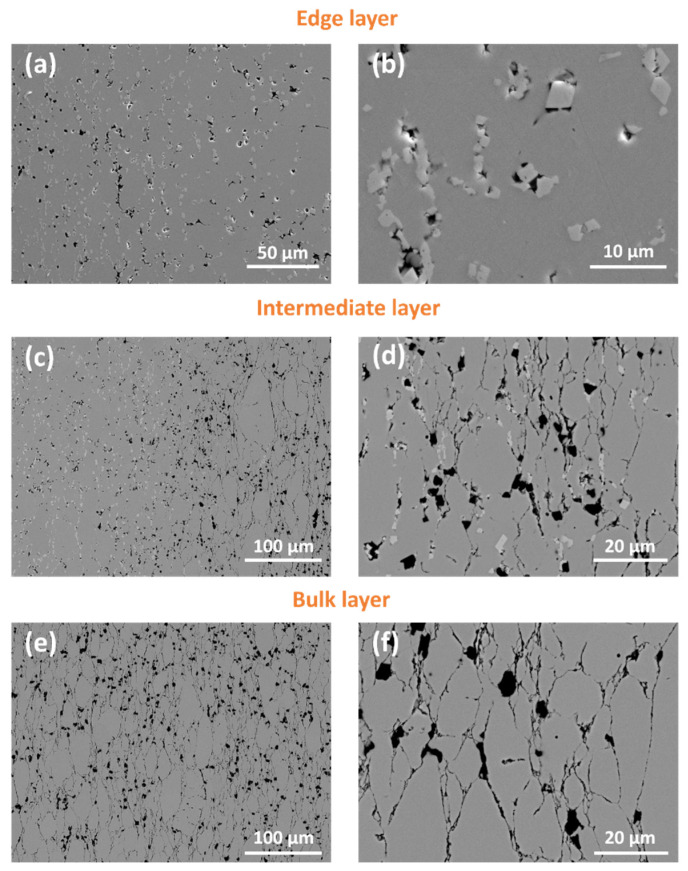
SEM images for 5 GAlN composite: (**a**,**b**)—edge layer, (**c**,**d**) intermediate layer; (**e**,**f**) bulk layer.

**Figure 10 materials-16-00618-f010:**
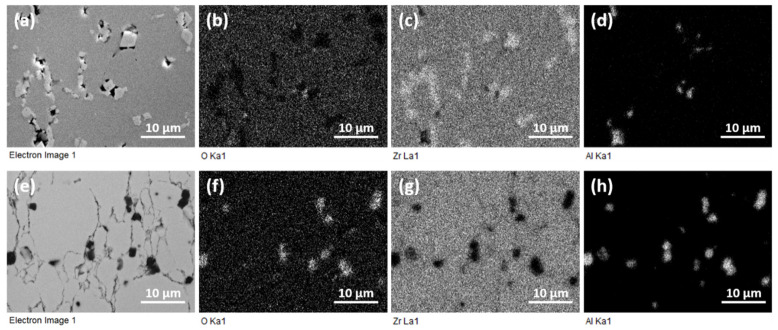
Elemental maps for 5 GAlN composite for edge (upper row) and bulk (bottom row) layers: (**a**,**e**)—SEM images; corresponding (**b**,**f**)—oxygen maps; (**c**,**g**)—zirconium maps; (**d**,**h**)—aluminum maps, taken from SEM image.

**Figure 11 materials-16-00618-f011:**
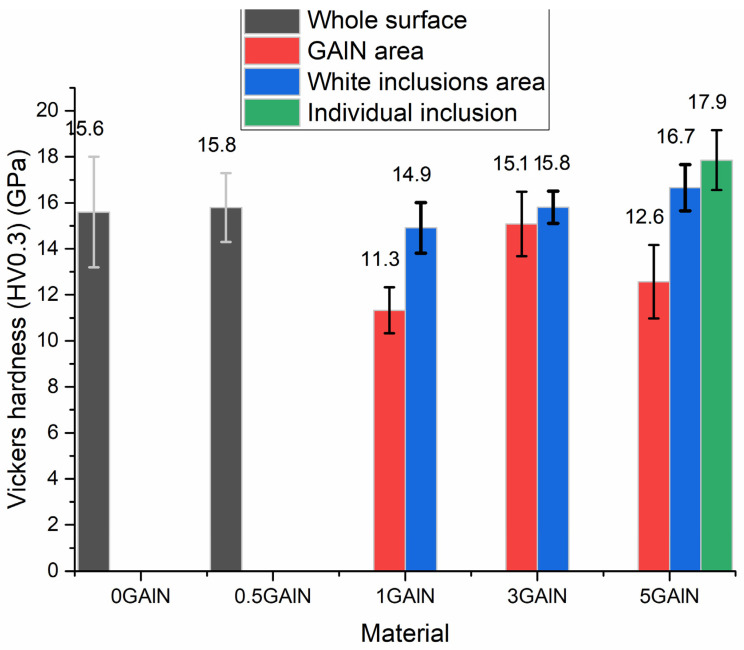
Vickers microhardness testing results, obtained for SPS-ed 0 GAlN ceramics and composites.

**Figure 12 materials-16-00618-f012:**
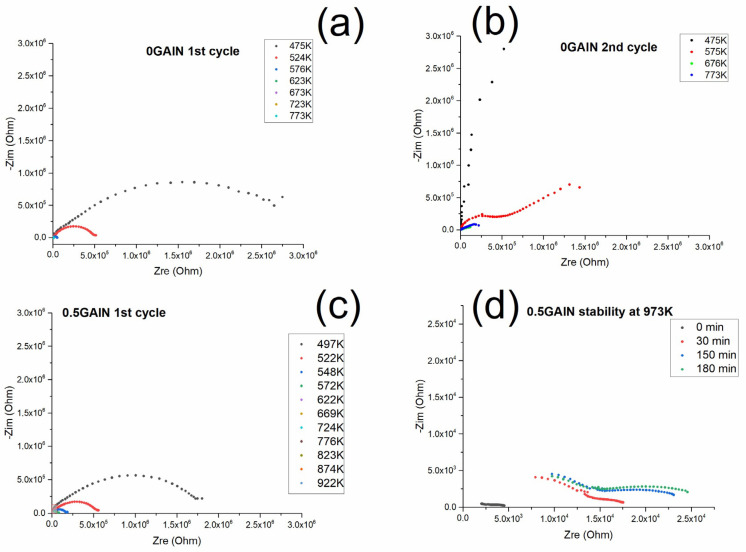
Electrochemical studies of (**a**) impedance spectra during the first cycle of heating/cooling for 0 GAlN; (**b**) impedance spectra during second cycle of heating/cooling for 0 GAlN (**c**) impedance spectra during the first cycle of heating/cooling for 0.5 GAlN; (**d**) impedance spectra at 973 K for 0.5 GAlN.

**Figure 13 materials-16-00618-f013:**
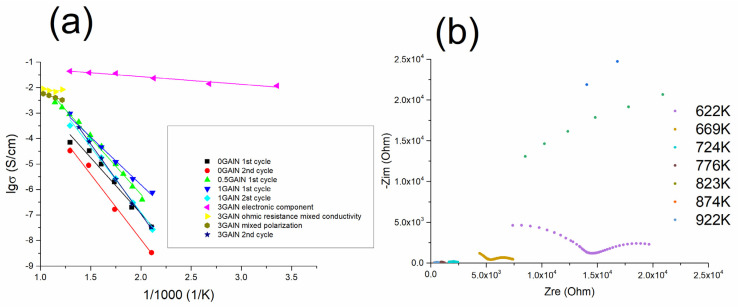
(**a**) Arrhenius plots of total conductivity for samples 0 GAlN–3 GAlN after the first and the second cycles of the thermal treatment; (**b**) the impedance spectra obtained at 622–922 K.

**Figure 14 materials-16-00618-f014:**
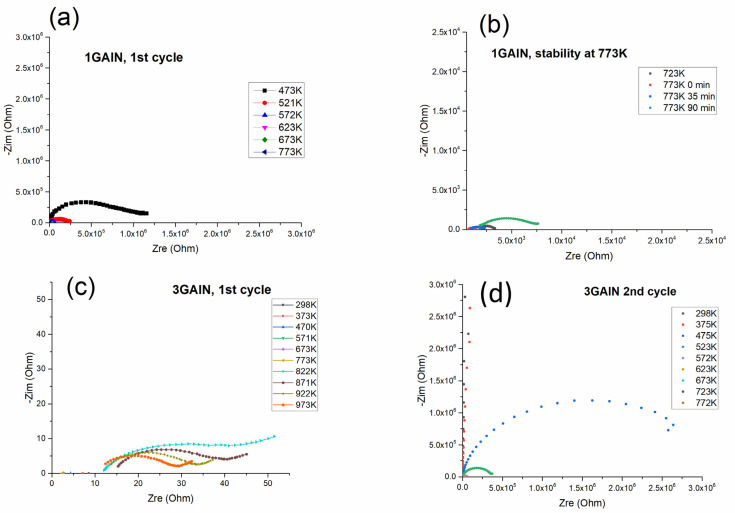
(**a**) impedance spectra for 1 GAlN during the first cycle of heating/cooling, (**b**) evolution of impedance spectra for 1 GAlN at 723 and 773 K; (**c**) impedance spectra for 3 GAlN during the first cycle of heating/cooling; (**d**) impedance spectra for 3 GAlN during the second cycle of heating/cooling.

**Figure 15 materials-16-00618-f015:**
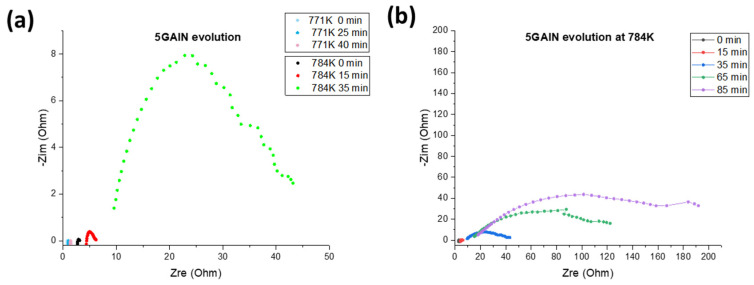
Evolution of the conductivity for 5 GAlN sample at (**a**) 771 K, (**b**) 784 K.

**Figure 16 materials-16-00618-f016:**
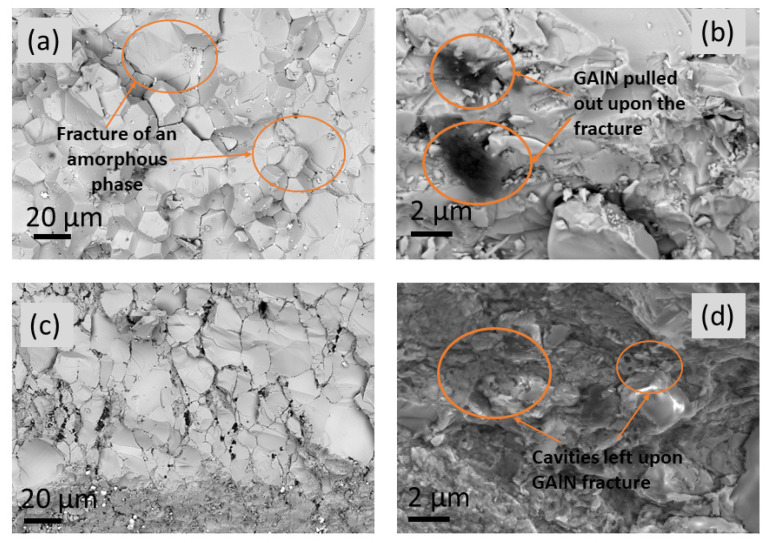
SEM photos of fracture surfaces, obtained for (**a**) 0 GAlN ceramics; (**b**) 1 GAlN composite; (**c**) 3 GAlN composite and (**d**) 5 GAlN composite.

**Figure 17 materials-16-00618-f017:**
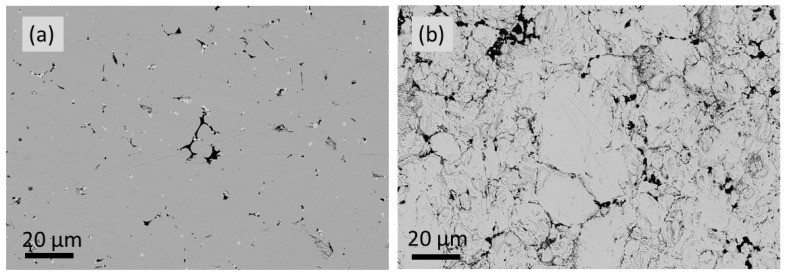
The microstructures of (**a**) 0.5 GAlN and (**b**) 3 GAlN composites after thermal cycling.

**Table 1 materials-16-00618-t001:** Obtained YSZ-GAlN samples labelling, their brutto compositions and used sintering conditions.

Sample Label	0 GAlN	0.5 GAlN	1 GAlN	3 GAlN	5 GAlN
Brutto-composition	91ZrO_2–_9Y_2_O_3_ (mol.%, YSZ)	99.5 wt.% YSZ + 0.5 wt.% GAlN	99 wt.% YSZ + 1 wt.% GAlN	97 wt.% YSZ + 3 wt.% GAlN	95 wt.% YSZ + 5 wt.% GAlN

**Table 2 materials-16-00618-t002:** Relative density, crystallinity, crystallite size and I_D_/I_G,_ and I_2D_/I_G_ intensity ratios of GAlN/YSZ composites.

Label	Relative Density, %	Crystallinity, %	Average Crystallite Size *d_cryst_*, nm	I_D_/I_G_	I_2D_/I_G_
0 GAlN	99.4 ± 0.47	66	87	-	-
0.5 GAlN	99.2 ± 0.07	69	89	-	-
1 GAlN	99.4 ± 0.03	75	94	0.78	0.76
3 GAlN	98.7 ± 0.01	77	97	0.79	0.87
5 GAlN	98.9 ± 0.02	84	104	0.62	0.57

**Table 3 materials-16-00618-t003:** Activation energies, calculated from Arrhenius plots for 0 GAlN–3 GAlN composites. No data were obtained for 5 GAlN due to the absence of the linear fitting.

Sample	0 GAlN	0.5 GAlN	1 GAlN	3 GAlN
Activation energy, eV First cycle	0.99	Total: 0.88 Grain boundaries 0.77 Grains 0.88	0.74	Electronic component: 0.06 ohmic component: 0.2 polarization component: 0.26
Activation energy, eV Second cycle	1.01	1.11	1.11	1.1

**Table 4 materials-16-00618-t004:** The conductivity of 5 GAlN composite at different temperatures and thermal evolution of conductivity at 771 K and 784 K.

Temperature, K Conductivity, S/cm		771 K		784 K	
		Time, min	Conductivity, S/cm	Time, min	Conductivity, S/cm
469	0.127	0	0.127	0	0.037
571	0.141	25	0.1	15	0.023
673	0.137	40	0.0709	35	0.011
720	0.132			65	0.0073
771	0.125			85	0.0062
798	0.0361				
cooling to 298	2 × 10^−8^				

## Data Availability

The research data is available upon request.

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
