# Peer review of "Mixed Electronic-Ionic Conductivity and Stability of Spark Plasma Sintered Graphene-Augmented Alumina Nanofibres Doped Yttria Stabilized Zirconia GAlN/YSZ Composites"

_materials, 2023, doi:10.3390/ma16020618_

Round 1
Reviewer 1 Report
The paper presents Experimental comparative study Mixed electronic-ionic conductivity and stability of spark 2 plasma sintered graphene-augmented alumina nanofibres 3 doped yttria stabilized zirconia GAlN/YSZ composites
This paper’s subject matter is well within the journal topic areas, however there are a number of problems and uncertainties that need the authors’ serious attention, and a significant re-write is required before we can assess it again. The reviewer recommends the minor revisions of paper befor acceptance. The following are problem areas:
1. The manuscript has many typos/errors (there also seems to be many spaces between words missing?).
2. Introduction it’s insufficient a more detail is required.
3. When talking about the materials and methods, the authors should clarify the reason for selecting the compositions that were chosen for their work.
4. Some Figures needs labelling properly.
5. Please provides a high resolution figures to meet the journal requirement
6. Discussion part is too weak,
7. Conclusion: It is too bulky. Make it concise form possibly with some numerical results.
8. Conclusions must be comprehensive and not written like a report.
Author Response
Reviewer 1
The paper presents Experimental comparative study Mixed electronic-ionic conductivity and stability of spark 2 plasma sintered graphene-augmented alumina nanofibres 3 doped yttria stabilized zirconia GAlN/YSZ composites
This paper’s subject matter is well within the journal topic areas, however there are a number of problems and uncertainties that need the authors’ serious attention, and a significant re-write is required before we can assess it again. The reviewer recommends the minor revisions of paper befor acceptance. The following are problem areas:
- The manuscript has many typos/errors (there also seems to be many spaces between words missing?).
Thank you very much for a comment and detailed reading of the manuscript. The errors and typos were fixed. I believe that the spaces between words are missing due to the automatic extraction of the text from Word file to the susy template. Lets hope the journal will fix this problem.
- Introduction it’s insufficient a more detail is required.
Thank you for the suggestion more details and references on the works devoted YSZ/graphene composites as well as graphene augmented alumina fibres were added.
- When talking about the materials and methods, the authors should clarify the reason for selecting the compositions that were chosen for their work.
Thank you for a comment. The compositions were chosen based on the literature data on the optimal amount of graphene additive to achieve the percolation threshold in zirconia-graphene composites [10,18,19,21,37, 40]..
Depending on the derivative type and ceramic matrix it varies from 1 to 5 wt.% of graphene additive. When rGO or GO were added to ceramic matrix the percolation threshold was achieved at 1-3 wt.% additive. In case of the tetragonal zirconia matrix and graphene augmented alumina fibres, the percolation threshold was achieved at 5 wt.% GAlN content [37]. A number of composites having different compositions (0.5-5 wt.% GAlN) were chosen to track the evolution of the structure and conductivity of composites and find the optimal additive amount for balanced electronic and ionic conductivity. The details were added to a manuscript.
- Some Figures needs labelling properly.
Thank you for the comment. That was corrected.
- Please provides a high resolution figures to meet the journal requirement
Thank you, the figures are now submitted along the revised version of the manuscript.
- Discussion part is too weak,
Thank you, in order to enhance the manuscript, the experimental part was changed according to reviewer#2 suggestion, the discussion was changed and strengthened. SEM photos of fracture were added to prove the hypothesis on microstructure evolution.
- Conclusion: It is too bulky. Make it concise form possibly with some numerical results.
Thank you. It was done.
- Conclusions must be comprehensive and not written like a report.
Thank you. That was changed.

Reviewer 2 Report
The present work addresses the study of the microstructure and the electrochemical behaviour of zirconia composites with graphene-augmented alumina nanofibers. The synthesis of the zirconia powders and the composites fabrication by Spark Plasma Sintering were also accomplished by the authors.
The fabrication of the composites is well described, along with its characterization, and the electrochemical tests seem to have been well-conducted. This manuscript will add to the increasing knowledge in the field, and should be considered for publication on Materials. However, there are some aspects that should be addressed, along with other improvements.
Abstract.
The last sentence of the abstract should be removed: “The results showed potential of the obtained zirconia-based composites as solid electrolyte membranes for intermediate temperatures”. The study clearly shows that the composites could not stand the thermal cycling, as they were degraded or even destroyed. Thermal cycling is essential for application as solid electrolyte.
Introduction.
Page 2, lines 71-77. Comments of the authors about rGO being the most commonly used Graphene derivative. More references about ceramic matrix composites with rGO as second phase should be included. Right now just a paper about graphene-reinforced aluminum matrix composites in mentioned (Ref 25). Moreover, in literature some authors have described the in-situ reduction of rGO during Spark Plasma Sintering of zirconia composites, which is one of the suggested key points to describe the enhanced bonding between the matrix and the additive, responsible of the improvement in the composite properties. See for example:
Enhancing the electrical conductivity of in-situ reduced graphene oxide-zirconia composites through the control of the processing routine. López-Pernía et al. Ceram. Int.
· The authors described the parameter “Crystallinity” ranging from 66 to 84%, with increasing values when increasing the nanofibers content (Table 2). Does it mean that the samples are partially amorphous? In that case, shouldn´t an amorphous halo be detected in the XRD diffractograms and the Raman spectra?
Were the synthesized YSZ powders amorphous? XRD of the powder should be included in the paper.
Why do the alumina fibers enhance the crystallinity?
If the samples present some percentage of amorphous phase the authors should include in the paper SEM images of fracture surface of the samples where this amorphous phase could be detected, and describe it.
Does the amorphous phase have an effect on electrical conductivity, It is not mentioned in the discussion.
· The authors related the presence of XRD peaks corresponding the ZrC phase to the reaction between the composite powder and the graphite die during sintering (page 4, lines 178-180). However, the sample with 5% nanofibers is the one with the highest amount of ZrC phase. Is it possible that the Graphene covering the nanofibers is also having a reaction with the zirconia powder during sintering?.
· Regarding the crystallite size, the non-linear change with the increase in fibers content was related to the presence of ZrC phase (page 6, lines 198-204). However, the authors have previously shown that polishing results in ZrC removal from the surface (Figure 2). So, calculations to obtain the crystallite size should be performed on the polished surfaces in order to avoid the effect of the ZrC phase on the surface of the samples.
· Figure 3(b) presents the zoomed area of the Raman spectra where the peaks corresponding to the cubic stabilized zirconia appear. A difference in the intensity ratio between the peaks at 610 and 680 cm-1 is observed between the composites with 0.5 GAlN and the ones with 1, 3 and 5 GAlN: the peak at 680 cm-1 is much lower in the latter. What is the reason for this effect?.
· ID/IG and I2D/IG ratios for each composite, and comments about them, should be included in the paper.
· Figure 4 and comments about it (page 7, lines 220-226) should be removed from the paper or moved to Supplementary Information. The conclusions from this Figure are very similar to the achieved ones from XRD analysis, and the paper is right now innecesarily long.
· The authors only included SEM images of the polished surfaces of the composites. Thus, the estimations for the ceramic grain size are not accurate. In order to reach conclusions about grain size or grain refinement with increasing the nanofibers content, analysis of the polished and revealed surfaces of the composites -where the grains can be precisely distinguished- should be addressed. The equivalent planar diameter should be used to estimate the ceramic grain size. The shape factor of the grains should be calculated in order to describe elongated grains. At least 500 grains should be analyzed for each composite.
· Fracture surface SEM images revealing the Graphene-alumina nanofibers are lacking in the paper, and should be included.
Mechanical testing.
· The microhardness tests were performed on the samples surfaces, and an effect of the ZrC phase was described (page 12, lines 304-311). Did the authors performe the indentations on polished surfaces? They have shown that by polishing, the surface ZrC phase can be removed, so it would be interesting to analyze the differences between unpolished/polished surfaces. This analysis should be included in the paper.
Electrochemical studies.
The electrochemical tests seem to have been well-conducted, however, the presentation of the data is quite confusing, the figures are not clear and the discussion is quite unclear. I suggest a restructuring of this section.
· The authors have presented impedance arcs in a range of temperatures for four samples (Figures 12-15). However, some arcs are so small that it is impossible to notice anything. Moreover, the authors repeatedly describe the impedance complex plane as presenting two arcs, but the two arcs are not detected in most of the cases. I suggest to select one or two of the temperatures and represent the arcs for the four samples just in one figure to illustrate the differences between them. The figures that are presented right now in the paper could be moved to Supplementary Information.
· Whereas for samples 0GAlN, 1 GAlN, and 3 GAlN Arrhenius plots of total conductivity in first and second cycle were presented, for the sample 0.5 GAlN Arrhenius plot of grains and grain boundary are presented. Why this difference in the presentation of the data?? It would be very interesting to present the Arrhenius plots of all the samples in the same figure, in order to allow an easy comparison.
· The sample 0.5 GAlN is finally destroyed after the measurements. This is clearly a problem of a non totally optimized processing of the sample. Did the authors observe cracks or failure in the other samples? It should be mentioned in the text.
· Why is not presented the Arrhenius plot for 5 GAlN sample?.
· A Table with the activation energies for all the samples would be very useful.
Discussion.
· Page 17, lines 434-436: “Strong interfacial bonding between zirconia matrix and GAlN, likely, favors to create faster pathway for a charge transfer on the surface of grains and, thus, the enhanced conductivity of grain boundaries”. It is not proved by the present work that there is a strong interfacial bonding between the matrix and the second phase. Authors must improve their analyses and characterizations of their samples (TEM, as an example) before affirming this.
· Page 18, lines 466-469: “rather small additions of AlGN result in the microstructure refinement …and thus allow to decrease the potential barrier for charge carrier to overcome for a charge transfer both though the bulk of composite and along the grains boundaries”. The authors should support this assumption on the literature. Please, add references about the effect of the microstructure refinement on the conductivity.
· The authors related the decrease in conductivity in the second thermal cycle to modifications in the crystallinity of the samples and grain growth (page 18). However, crystallinity data or microstructure after the thermal cycling are not presented in the paper, so this assumption is highly speculative. If the authors wish to maintain this conclusion, the characterization of the samples after thermal cycling should be completed. Anyway, it looks to me that the decrease in conductivity is related to a clear degradation of graphene during the measurements. What is the reason for this degradation? As the measurements were performed in Argon, it is quite surprising that the graphene is degraded.
· Page 19, lines 516-517: “…make the composite attractive as material for solid oxide fuel cells and oxygen pumps”. This closing sentence should be removed. The graphene in the samples is clearly degraded after thermal cycling, the sample 0.5 GAlN is even destroyed. This makes the samples not suitable for applications.
Conclusions.
This Section should be rewritten according to the modifications in the Discussion section.
Figures.
· Figures 1-4: numbers in the y-axis should be removed. Numbers in x-axis and axis Titles should be bigger.
· SEM figures: use the same contrast in all of them.
· Arrhenius plots: use the same scale for all of them.
Author Response
The detailed answer to the reviewer’s comments
Dear reviewer! Thank you for your time and the possibility to review the manuscript promptly. We made the changes according to the reviewers’ suggestions, added new experimental data, changed the discussion and conclusions. We believe that it allowed to enhance the manuscript and the hypothesis made and now the manuscript is ready for the publication. All the changes are highlighted in yellow. Let us answer to all comments and questions in detail.
Reviewer 1
The present work addresses the study of the microstructure and the electrochemical behaviour of zirconia composites with graphene-augmented alumina nanofibers. The synthesis of the zirconia powders and the composites fabrication by Spark Plasma Sintering were also accomplished by the authors.
The fabrication of the composites is well described, along with its characterization, and the electrochemical tests seem to have been well-conducted. This manuscript will add to the increasing knowledge in the field, and should be considered for publication on Materials. However, there are some aspects that should be addressed, along with other improvements.
Abstract.
The last sentence of the abstract should be removed: “The results showed potential of the obtained zirconia-based composites as solid electrolyte membranes for intermediate temperatures”. The study clearly shows that the composites could not stand the thermal cycling, as they were degraded or even destroyed. Thermal cycling is essential for application as solid electrolyte.
I agree, it was deleted.
Introduction.
Page 2, lines 71-77. Comments of the authors about rGO being the most commonly used Graphene derivative. More references about ceramic matrix composites with rGO as second phase should be included. Right now just a paper about graphene-reinforced aluminum matrix composites in mentioned (Ref 25). Moreover, in literature some authors have described the in-situ reduction of rGO during Spark Plasma Sintering of zirconia composites, which is one of the suggested key points to describe the enhanced bonding between the matrix and the additive, responsible of the improvement in the composite properties. See for example:
Enhancing the electrical conductivity of in-situ reduced graphene oxide-zirconia composites through the control of the processing routine. López-Pernía et al. Ceram. Int. 47 (2021) 9382–9391.
Fabrication and properties of in situ reduced graphene oxide-toughened zirconia composite ceramics. Zeng et al. J. Am. Ceram. Soc. 101 (2018) 3498–3507.
Thank you for a suggestion. The references were added to the manuscript along several other references on rGO, GO, graphene nanoplatelets.
Experimental.
Table 1. The column “Sintering conditions” should be removed, as the information is already included in the text (Page 3, lines 128-131.
Thank you. It was done.
Details about how the temperature measurement was carried out during SPS should be included in the text.
With no doubt, the correct sintering temperature is the most important process parameter besides time and heating rate. The FCT Systeme machine allows precise temperature measurement and control. Due to a special design FAST/SPS systems are measuring the temperature in the vicinity of the powder compact center, which gives a much more significant value than the measurement of the die temperature. The details were included in the text.
Results.
Microstructure.
The authors described the parameter “Crystallinity” ranging from 66 to 84%, with increasing values when increasing the nanofibers content (Table 2). Does it mean that the samples are partially amorphous? In that case, shouldn´t an amorphous halo be detected in the XRD diffractograms and the Raman spectra?
Thank you for a question. The crystallinity is a ratio of crystalline to amorphous phase, which is determined using the standard software package of the diffractometer. The presence of an amorphous phase is shown by the peak broadening the XRD pattens in Fig.1. The amorphous halo is seen for amorphous substances or in case of the beginning of the crystallization. The crystallization takes place when the crystallinity degree reaches more than 23%. Upon higher crystallinity degree, the XRD pattern of the sample is referred to a crystalline material. In addition, there are no fully crystalline samples (100% crystallinity). The existence of the structural defects (both point and planar), phase of grain boundaries makes it impossible to reach 100% crystallinity of any material. The details were added into the text of the manuscript when describing Fig.1.
Were the synthesized YSZ powders amorphous? XRD of the powder should be included in the paper.
Yes, the powders after the synthesis were amorphous. The data was included in the manuscript in the synthesis section. Zirconia based precursor after annealing at 1073K has a crystallinity value of ~30%. The XRD patten was presented in our previous paper [42]. In order not to duplicate the data and not to make paper excessively long, we inserted just a reference for a paper.
Why do the alumina fibers enhance the crystallinity?
According to [33], graphene changes the mechanism of composite sintering. Based on dilatometry data zirconia composites demonstrated mixed mechanism of sintering with a grain boundary diffusion predominance. We believe, that similar mechanism takes place upon the GAlN introduction and results to the crystallinity enhancement. It was added into the discussion
If the samples present some percentage of amorphous phase the authors should include in the paper SEM images of fracture surface of the samples where this amorphous phase could be detected, and describe it.
Thank you for a suggestion. Corresponding SEM images of fracture were added for sample 0GAlN, where the presence of the amorphous phase is clearly seen. The structural features, referred to the fracture of the amorphous phase were described.
Does the amorphous phase have an effect on electrical conductivity, It is not mentioned in the discussion.
Certainly, better the crystal structure is formed, less structure defects are present, finer are the grain boundaries, lower is the resistance and, hence the conductivity increase is seen. We mentioned it in the revised discussion when talking about the conductivity and crystallinity of samples.
- The authors related the presence of XRD peaks corresponding the ZrC phase to the reaction between the composite powder and the graphite die during sintering (page 4, lines 178-180). However, the sample with 5% nanofibers is the one with the highest amount of ZrC phase. Is it possible that the Graphene covering the nanofibers is also having a reaction with the zirconia powder during sintering?
Yes, it might be possible since the small amounts of ZrC (less than XRD detection limit) are present in the bulk of composite Thank you for a comment. It was added into the discussion. However the main reason is still considered to be a diffusion gradient between a die and the composite since the edge is enriched both with ZrC and GAlN
- Regarding the crystallite size, the non-linear change with the increase in fibers content was related to the presence of ZrC phase (page 6, lines 198-204). However, the authors have previously shown that polishing results in ZrC removal from the surface (Figure 2). So, calculations to obtain the crystallite size should be performed on the polished surfaces in order to avoid the effect of the ZrC phase on the surface of the samples.
Thank you! The crystallite size was recalculated using the XRD patterns of the polished samples. Now it is do in accordance with crystallinity data.
- Figure 3(b) presents the zoomed area of the Raman spectra where the peaks corresponding to the cubic stabilized zirconia appear. A difference in the intensity ratio between the peaks at 610 and 680 cm-1is observed between the composites with 0.5 GAlN and the ones with 1, 3 and 5 GAlN: the peak at 680 cm-1 is much lower in the latter. What is the reason for this effect?.
If exclude the translation operation, no more than 3m-3 fundamental oscillations can be seen in the Raman spectrum. For cubic ZrO2 the number of fundamental vibrations is 33. In fact, due to the high symmetry of the structure, the vibrations deteriorate: the Raman spectrum of cubic ZrO2 A band about 610 cm-1 is a main band corresponding to the vibration of the O-Zr-O atoms. The band at 610 cm ' corresponds to the zone-center F2g mode which is the only expected first-order Raman-active mode for this crystal class. The band at 680 cm-1 is related to the disorder-induced scatterring. [https://doi.org/10.1103/PhysRevB.51.201]. So, the decrease of the intensity of the band at 680 cm-1 in spectra obtained indicates the structural ordering of the composites and is in accordance with the crystallinity and crystallite sizes increase in the composites. It was added into the results and discussion section.
- ID/IGand I2D/IG ratios for each composite, and comments about them, should be included in the paper.
Thank you. The ID/IG and I2D/IG intensities ratios were calculated to be less than 1 for 1, 3 and 5 GAlN composites, indicating that no damage of the graphene structure took place during sintering.
- Figure 4 and comments about it (page 7, lines 220-226) should be removed from the paper or moved to Supplementary Information. The conclusions from this Figure are very similar to the achieved ones from XRD analysis, and the paper is right now innecesarily long.
Thank you. It was done.
- The authors only included SEM images of the polished surfaces of the composites. Thus, the estimations for the ceramic grain size are not accurate. In order to reach conclusions about grain size or grain refinement with increasing the nanofibers content, analysis of the polished and revealed surfaces of the composites -where the grains can be precisely distinguished- should be addressed. The equivalent planar diameter should be used to estimate the ceramic grain size. The shape factor of the grains should be calculated in order to describe elongated grains. At least 500 grains should be analyzed for each composite.
Thank you for the suggestion. We tried to do it. However now there was no technical possibility to estimate the grain sizes. So the data about grain sizes estimation was removed from the manuscript.
- Fracture surface SEM images revealing the Graphene-alumina nanofibers are lacking in the paper, and should be included.
Thank you. Corresponding SEM data was included into the manuscript.
Mechanical testing.
- The microhardness tests were performed on the sample surfaces, and an effect of the ZrC phase was described (page 12, lines 304-311). Did the authors performe the indentations on polished surfaces? They have shown that by polishing, the surface ZrC phase can be removed, so it would be interesting to analyze the differences between unpolished/polished surfaces. This analysis should be included in the paper.
The indentation was made for polished samples. All samples that we currently have, are polished. There is no possibility to perform testing and manufacture new samples for testing since it was done in Estonia and because of political situation the collaboration is stopped.
In general, the presence of ZrC on the surface would certainly lead to overall microhardness increase. At the same time surface roughness of the unpolished samples will introduce the additional statistical errors and the increased error of the experiment. So generally, the analysis of differences between unpolished/polished samples seems to be not significantly important.
Electrochemical studies.
The electrochemical tests seem to have been well-conducted, however, the presentation of the data is quite confusing, the figures are not clear and the discussion is quite unclear. I suggest a restructuring of this section.
- The authors have presented impedance arcs in a range of temperatures for four samples (Figures 12-15). However, some arcs are so small that it is impossible to notice anything. Moreover, the authors repeatedly describe the impedance complex plane as presenting two arcs, but the two arcs are not detected in most of the cases. I suggest to select one or two of the temperatures and represent the arcs for the four samples just in one figure to illustrate the differences between them. The figures that are presented right now in the paper could be moved to Supplementary Information.
Thank you for a suggestion, indeed the standard temperature step of measurement did not allow to detect all full semiarches at all temperatures. It is seen that one arch is ended and another starts. In order to make it more clear for a reader the description was changes and more details were added. However if put all spectra together it will be very difficult to understand the kinetic measurement for each sample. We decided to keep it as it is with more clarified description but put all Arrhenious plots in the one Figure.
- Whereas for samples 0GAlN, 1 GAlN, and 3 GAlN Arrhenius plots of total conductivity in first and second cycle were presented, for the sample 0.5 GAlN Arrhenius plot of grains and grain boundary are presented. Why this difference in the presentation of the data?? It would be very interesting to present the Arrhenius plots of all the samples in the same figure, in order to allow an easy comparison.
Thank you, it was done since for 0.5 GAlN the determination of the grain and grain boundaries impacts was possible. For other composites it was not possible to distinguish the components that well. In order to make it clear the total conductivities were presented for all samples in the same figure.
- The sample 0.5 GAlN is finally destroyed after the measurements. This is clearly a problem of a non totally optimized processing of the sample. Did the authors observe cracks or failure in the other samples? It should be mentioned in the text.
Only 0.5GAlN was cracked after electrochemical measurements.The conditions of sintering were optimized before by I. Hussainova group for composites with GAlN doped with tetragonal zirconia matrix. So the same SPS conditions were used here. We believe that the addition of 0.5 wt.% GAlN somehow induced thermal stresses in the composite and induced cracking. So the composition was not optimal. All the other composite were stable and did not crack.
- Why is not presented the Arrhenius plot for 5 GAlN sample?
The conductivity of 5 GAlN sample with temperature does not fit the Arrhenius dependence. That is why it was not included. It was mentioned in the text.
- A Table with the activation energies for all the samples would be very useful.
Thank you. Indeed, the table was added.
Discussion.
- Page 17, lines 434-436: “Strong interfacial bonding between zirconia matrix and GAlN, likely, favors to create faster pathway for a charge transfer on the surface of grains and, thus, the enhanced conductivity of grain boundaries”. It is not proved by the present work that there is a strong interfacial bonding between the matrix and the second phase. Authors must improve their analyses and characterizations of their samples (TEM, as an example) before affirming this.
Yes, I agree and always suggest the same when being a reviewer. Unfortunately, there is no technical possibility now to perform TEM measurements. At the same time in the previous publications of prof. Hussainova group, corresponding measurements were performed for GAlN-tetragonal zirconia matrix and the strong interfacial bonding between the zirconia matrix and the GAlN phase was confirmed. The discussion was changed and the corresponding literature references were added.
- Page 18, lines 466-469: “rather small additions of AlGN result in the microstructure refinement …and thus allow to decrease the potential barrier for charge carrier to overcome for a charge transfer both though the bulk of composite and along the grains boundaries”. The authors should support this assumption on the literature. Please, add references about the effect of the microstructure refinement on the conductivity.
Thank you. It is a good point
- The authors related the decrease in conductivity in the second thermal cycle to modifications in the crystallinity of the samples and grain growth (page 18). However, crystallinity data or microstructure after the thermal cycling are not presented in the paper, so this assumption is highly speculative. If the authors wish to maintain this conclusion, the characterization of the samples after thermal cycling should be completed. Anyway, it looks to me that the decrease in conductivity is related to a clear degradation of graphene during the measurements. What is the reason for this degradation? As the measurements were performed in Argon, it is quite surprising that the graphene is degraded.
Thank you, the SEM data of the composites after the thermal cycling was added to the manuscript along with crystallinity to support the conclusions. Indeed, for 3GAIN the grains change their shape from elongated to the typical for cublic zirconia ceramics being the reason of break down of the GAlN fibres. About the degrading, STA data in Supplementary confirms the degrading of graphene. It is likely due to a very small amount of graphene on the surface of alumina that present in the composites. For 3GAIN and 5GAIN it is just 0.3 and 0.5 wt.% Gr. Additionally, in the previous publications considering GAlN composites, the mechanical testing and thermal conductivity measurements were performed at room temperature.
- Page 19, lines 516-517: “…make the composite attractive as material for solid oxide fuel cells and oxygen pumps”. This closing sentence should be removed. The graphene in the samples is clearly degraded after thermal cycling, the sample 0.5 GAlN is even destroyed. This makes the samples not suitable for applications.
Thank you, it was removed.
Conclusions.
This Section should be rewritten according to the modifications in the Discussion section.
Thank you Done.
Figures.
- Figures 1-4: numbers in the y-axis should be removed. Numbers in x-axis and axis Titles should be bigger.
Thank you, it was done.
- SEM figures: use the same contrast in all of them.
The contrast depends on the type of the specimen and the tasks of the measurement. So it is impossible to use the same contrast for all of them
- Arrhenius plots: use the same scale for all of them.

Reviewer 3 Report
This paper fabricated composites of GAIN-YSZ by spark plasma sintering. And the evolution of conductivity and hardness was investigated. I have some minor comments before this work can be published.
1. In Figure 1, why there is no ZrC phase for the sample of 0.5GAIN and 3GAIN.
2. What is the effect or function of polishing?
Author Response
Dear reviewer! Thank you for your time and the possibility to review the manuscript promptly. We made the changes according to the reviewers’ suggestions, added new experimental data, changed the discussion and conclusions. We believe that it allowed to enhance the manuscript and the hypothesis made and now the manuscript is ready for the publication. All the changes are highlighted in yellow. Let us answer to all comments and questions in detail.
Comments and Suggestions for Authors
This paper fabricated composites of GAIN-YSZ by spark plasma sintering. And the evolution of conductivity and hardness was investigated. I have some minor comments before this work can be published.
- In Figure 1, why there is no ZrC phase for the sample of 0.5GAIN and 3GAIN.
Thank you, ZrC is present in all the samples if take a closer look at the SEM figures. In case of 0.5GAIN and 3GAIN its limit is under the XRD detection.
- What is the effect or function of polishing?
Thank you for a question, polishing is a standard procedure to prepare the surface of mechanical testing or SEM and EDS data. It is done to avoid the impact of the roughness and avoid experimental errors and reveal grains and some other microstructure features. In our case polishing was necessary to remove the carbon layer formed on the composite surface after sintering in the graphite die.

Round 2
Reviewer 2 Report
The authors fulfilled all my suggestions and the paper is greatly improved.
I consider it should be accepted for publication.